# A Matter of Health? A 24-Week Daily and Weekly Diary Study on Workplace Bullying Perpetrators’ Psychological and Physical Health

**DOI:** 10.3390/ijerph20010479

**Published:** 2022-12-28

**Authors:** Gülüm Özer, Yannick Griep, Jordi Escartín

**Affiliations:** 1Department of Social Psychology and Quantitative Psychology, University of Barcelona, 08035 Barcelona, Spain; 2Behavioural Science Institute, Radboud University, 6525 GD Nijmegen, The Netherlands; 3Stress Research Institute, Stockholm University, 16407 Stockholm, Sweden; 4Institute of Psychiatry, Psychology & Neuroscience, King’s College London, London SE5 8AF, UK

**Keywords:** workplace bullying perpetration, daily and weekly diary study, being bullied, physical activity, sleep, fitness trackers

## Abstract

Workplace bullying (WB) studies focusing on perpetrators are increasing. Many processes, events, circumstances and individual states are being studied to understand and inhibit what causes some employees to become perpetrators. Using a 24-week diary design and drawing on the Conservation of Resources Theory, we investigated how sleep, physical activity (PA), and being bullied predicted perpetration on a within-level. On a between-level, we controlled for a supervisory position, psychological distress and mental illnesses over 38 employees from Spain and Turkey. Their average age was 38.84 years (*SD* = 11.75). They were from diverse sectors (15.8% in manufacturing, 15.8% in education, 13.2% in wholesale and retail trade, 13.2% in information and communication, 7.9% in health, 7.9% in other services and 26.3% from other sectors) with diverse professions such as finance manager, psychologist, graphic designer, academic, human resources professional, forensic doctor, IT and Administration head, municipality admin executive, waiter, and sales executives. Data collection was conducted over 24 consecutive work weeks, where only 31 participants were involved in perpetration (final observations = 720). We analyzed the data using multilevel structural equation modeling decomposed into within-and-between-person variance parts. The results indicated that on a within-level, PA as steps taken during the work week and being bullied positively predicted perpetration the same week, while sleep quality did not. By connecting sleep, physical exercise and WB literature, we draw attention to the health condition of perpetrators. Organizations should actively inhibit workplace bullying and be mindful of employees’ physical activities at work or commuting to work. Managers should also be attentive to physical fatigue that employees may feel due to their responsibilities in their private lives and allow employees to rest and recuperate to inhibit negative behaviors at work.

## 1. Introduction

Workplace bullying (WB) is a common and persistent phenomenon, defined as a severe and damaging interpersonal behavior [1] occurring regularly and repeatedly over time. Previous studies showed that the prevalence rates of WB perpetration vary between 2.8% [2], 4% [3], and 5% [4]. The impact of WB on targets stretches from sleep problems [5,6], and mental health issues [7] to extreme suicidal thoughts [8]; perpetrators are psychologically distressed [9] and suffer mental illnesses [10]. While we do not know much about perpetrators’ physical activities, a previous study showed that they had elevated adverse physical symptoms, including trouble sleeping, and tended to be supervisors rather than employees [10]. Previous research showed that targets tended to be high in neuroticism [11], and perpetrators tended to be aggressive [9], disagreeable [12] and less conscientious [13]. Cross-sectional research on the work environment found that WB perpetration is higher when organizational trust and justice are low [10]; when there is widespread inappropriate social conduct [14] and team fit is low [15]; when leaders are absent [16], or passive avoidant [17] and when employees are being bullied [18,19].

Research on WB perpetrators largely depends on between-person designs, limiting our knowledge of how fluctuations in exposure to stressors relate to within-level fluctuations of WB perpetration behavior. The present study focused on perpetrators’ well-being as antecedents of WB perpetration for 24 working weeks. We aimed to (i) monitor WB perpetration behavior, its frequency, intensity, and duration, and (ii) combine individual and work-related factors explaining how these antecedents may develop into WB perpetration. The three-way model [20] suggests that bullying perpetration may arise due to (i) inefficient coping with frustration, (ii) escalated conflicts and (iii) destructive team and organizational cultures or habits. We will also test if our results support the model where antecedents to perpetration may affect these processes.

## 2. Theoretical Background and Hypotheses

Conservation of Resources Theory (COR) [21,22] suggests that individuals seek to obtain, retain, protect, and cherish resources (e.g., tools for work, health, employment, tenure, critical skills, personal traits, and knowledge). Psychological stress occurs when individuals’ resources are threatened with loss, suffer actual loss, or when individuals fail to gain sufficient resources following significant resource investment. Individuals may become defensive, aggressive, and irrational to preserve the self, and WB perpetration may become a plausible option. Previously COR was used to explain perpetration as a coping mechanism that is activated due to loss of resources while experiencing undermining and verbal abuse [19]; task conflicts [23]; a stressful work environment and being bullied [18]. Therefore, we aim to study within-person exposure to bullying, and fluctuations in sleep and physical activities as factors that would drain or protect individuals and lead to subsequent within-level WB perpetration from the perspective of COR [24]. Our hypothesized model is in Figure 1.

### 2.1. Exposure to Bullying as a Predictor of WB Perpetration

Employees experiencing WB feel psychological distress [25], are depleted, and their personal resources are eroded [26]. According to COR, these negative feelings show that the individual has lost critical resources such as feeling successful and valuable [22]. To stop the resource drain and protect the self, individuals may act in anger and frustration and resort to bullying others as a coping and recovery strategy. If they feel they have recuperated the lost resources, they may get better at this strategy and use it as a long-term strategy [24], hence becoming perpetrators. Previously, cross-sectional [24] and longitudinal studies [27] showed that bullying correlates with and predicts WB perpetration. Therefore, based on previous studies, we argue the following.

**Hypothesis** **1.**
*Weekly exposure to bullying will positively predict weekly WB perpetration.*


### 2.2. Physical Activity as a Predictor of WB Perpetration

Physical activity (PA) is any bodily movement produced by skeletal muscle that requires energy, such as walking, cycling, lifting, dancing, and cleaning. It can improve mental health, quality of life and well-being if undertaken regularly and of sufficient duration and intensity [28]. Research showed that regular leisure-time PA is associated with higher work ability [29]; improves personal relations [30]; works as a stress buffer [31] and protects against job burnout and depression [32]. However, worktime and leisure-time physical activities do not provide similar health benefits. Studies showed that physical activity at work is positively related to exhaustion [33] and depressive symptoms [34]; it increases the risk of long-term sickness absence [35] and cardiovascular diseases [36]. While regular PA is widely accepted to be essential for longevity and health, the optimal dose of PA for a better life may depend on the individual circumstances (e.g., the overall health of the individual) and situational factors (e.g., activity being voluntary or not).

Previous research demonstrated how weekends help individuals to recover from stress and self-control difficulties [37]. On the other hand, individuals reported reduced resources from Monday to Friday, typically leading to impaired performance on self-control tasks [38]. Therefore, by distinguishing PA by weekdays and weekends, we explored the time dimension of PA and how PA impacts WB perpetration behavior for each individual. As all participants worked during the week, weekends provided them with natural breaks for recovery. We assumed the weekend PA would be leisure activities, energizing the individual and providing coping resources against work stress. Even with excesses over individuals’ average activities, weekend PA would inhibit WB perpetration in the coming days of the week. We assumed that weekday PA may be carried out in a combination of domains (e.g., work-related, commuting to work, household tasks, and leisure activities) where the total PA during the week may cause physical exhaustion and drain resources. Therefore, we wanted to test if, on a within-level, excess PA during the weekdays may cause physical exhaustion, drain resources, and trigger WB perpetration behavior.

**Hypothesis** **2a.**
*Weekly average PA (steps taken) during the week will positively predict weekly WB perpetration.*


**Hypothesis** **2b.**
*Weekly average PA (steps taken) during the weekend will negatively predict weekly WB perpetration.*


### 2.3. Sleep as a Predictor of WB Perpetration

Among the initial objective list of resources, “time for adequate sleep” was one of the resources in the COR [22]. Sleep quantity is the amount of time an individual spends in a sleeping state, and sleep quality is a combination of factors such as difficulty falling asleep and staying asleep. Lack of both dimensions of sleep affects the individual [39] as sleep is important for the recovery of physiological resources regulating self-control [40] and restoring resources [41]. Studies showed that poor sleep quality correlates with interpersonal conflict [42]; predicts increased reactive aggression [43] and frequent anger [44]. Diary studies showed that supervisors’ sleep quality predicts abusive supervision [45], interpersonal conflict and depleted feelings [38], unethical behavior, and social deviance [37] the next day. Sleep deprivation decreases individuals’ self-control [37], increases hostility, and increases workplace deviance, such as violence and interpersonal rudeness [46]. While the lack of sleep quality and duration correlate and predict negative behavior, short sleep duration did not significantly explain aggression [43], abusive supervision [45] and interpersonal conflict [38].

Therefore, when individuals are stressed, they suffer from sleep troubles and are drained of their resources. When resources are exhausted, they become defensive, strive to preserve themselves, and often act aggressively and irrationally in the short term. As a coping strategy, they may feel the urge to control others to feel in charge and belittle others to feel accomplished and successful. However, as such resource gains are perceived as smaller and slower than resource losses, balancing resources takes longer [24] and may lead to long-term, sustained negative acts, defined as WB perpetration. Therefore, based on previous research and the COR, we argue that lack of weekly sleep quality and duration will be related to the weekly WB perpetration.

**Hypothesis** **3a.**
*Weekly sleep quality will negatively predict weekly WB perpetration.*


**Hypothesis** **3b.**
*Weekly sleep duration will negatively predict weekly WB perpetration.*


## 3. Method

### 3.1. Participants

We conducted a three-wave longitudinal and a diary study simultaneously measuring different antecedents for perpetration. The results of the first wave longitudinal study (LS) (*N* = 2508) were used to recruit participants and to set control variables for the diary study. For the LS, we collected data mainly by reaching out to two psychology and organizational psychology professors at Spanish and Turkish universities. We invited them to help in the initial wave data-gathering phase by encouraging their students to find respondents who worked at least eight hours per week, in line with the ILO definition of being employed [47], in exchange for extra credit. Data obtained by students gathering respondents were heterogeneous and thus more likely to be generalizable [48]. Respondents were informed that the study was about employee health without explaining the hypotheses and disguising that it was a study on WB. Data were collected via the Qualtrics survey tool. The study was conducted under the approval of the Bioethical Committee of the University of Barcelona covering the countries mentioned (protocol code IRB00003099, approved as of 5 October 2020). All respondents provided electronic informed consent to participate, their data to be used for publication and entered their email addresses to be further contacted for research.

The following criteria were used to invite first-wave respondents to the diary study (i) bullies and perpetrators, (ii) victims and targets, since being a target strongly predicts being a perpetrator, (iii) participants high in neuroticism, low in agreeableness and conscientiousness as these traits are related to WB bullying and perpetration. Therefore, from the LS T1 results, employees with higher-than-average perpetration (*M =* 1.22, *SD =* .57), target (*M* = 1.66, *SD* = .91), victim (*M* = 1.47, *SD* = 1.15), and bullying scores (*M* = 1.14, *SD* = .66) were invited to the diary study. Additionally, employees with lower-than-average conscientiousness (*M* = 5.43, *SD* = 1.17), agreeableness (*M* = 5.52, *SD* = 1.05), and higher-than-average neuroticism (*M* = 3.74, *SD* = 1.15) traits were also invited to join the diary study. Therefore, 493 participants were invited to the study disguised as wellness training called the “Leadership Wellness Program”. Thirty-eight individuals adhering to our inclusion criteria and who were currently working joined the diary study. Seven participants reported no perpetration during the diary study. Therefore, they were taken out from the analysis leaving a total of 31 participants (Table 1). All the analyses conducted on the within-level were based on the 31 participants involved in WB perpetration during the diary study, while between-level analysis is based on their scores at the first wave of LS. The second (T2) and third waves (T3) of the LS are not part of the present study.

The participants provided 720 observations, where 28 participants completed 24 waves, one participant 19 waves, one participant 15 waves and one participant 14 waves of data collection. We observed WB perpetration 720 times of the possible 912 (24 weeks × 38 participants), yielding an observation rate of 78.95%.

Their average age was 37.94 (*SD* = 12.27), with 6.39 years of tenure in work-life. On average, they worked 5.10 days (*SD* = .70) weekly. Participants had various professions (e.g., academicians, customer support and sales representatives, finance managers, medical doctors, graphic designers, human resources professionals, IT managers, municipality administration officials, and teachers). They were from various sectors: manufacturing (16.67%), wholesale and retail trade (16.67%), information and communication (16.67%), education (13.33%), health (10.00%), and other sectors (20.00%). Of the 31 participants, only four participants (12.90%) worked in a gender-balanced environment, twelve (38.71%) were supervisors, and eighteen (58.06%) were living in Turkey. Fourteen participants (45.16%) were female, and 9 (29.03%) reported having or had been diagnosed with a mental illness (Mental Illness score was formed by (i) asking for a “yes = 1” or “no = 0” answer to; “Has a physician ever informed you that you have or have had chronic diseases listed below?” Depression [49]. (ii) Open-ended question on other chronic illnesses. Then, mental illnesses mentioned (e.g., bipolar disorder, obsessive-compulsive disorder) were scored as “yes = 1” in the other mental illnesses column. Finally, depression scores and other mental illnesses columns were combined to form “mental illnesses”). Participants were not subject to COVID-19 lockdown measures. Based on the first wave of LS, they scored 4.05 (*SD* = 1.17) on extraversion (Personality traits were measured using the 20-item mini-IPIP scale [50], (1 = completely false, 7= completely true)), 5.65 (*SD* = .83) on agreeableness, 5.21 (*SD* = 1.15) on conscientiousness, 4.35 (*SD* = 1.30) on neuroticism, 4.92 (*SD* = 1.14) on intelligence and imagination. They reported that they were victims of bullying (T1 Victim score was measured by single-item questions with a bullying definition (COPSOQ III); “Bullying means that a person repeatedly is exposed to unpleasant or degrading treatment, and that the person finds it difficult to defend himself or herself against it”. Have you been exposed to bullying at your workplace in the last 6 months? (1 = never, 7 = very frequently) [51]). (*M* = 2.16, *SD* = 1.70), and they bullied others (T1 Bully question was obtained by modifying the bullying questions into an active form; “Have you bullied others at your workplace in the last 6 months? (1 = never, 7 = very frequently).) in their current jobs (*M* = 1.32, *SD* = 1.14). We also inquired about bullying experiences and WB perpetration through a behavioral approach. The participants scored 2.28 (*SD* = .86) on target questions (Workplace bullying was measured by a 4-item EAPA-T-R scale [52] (1 = never, 7 = very frequently/more than once a week). Workplace bullying perpetration was measured by the same scale by adopting the questions to an active format), 1.30 (*SD* = .39) on WB perpetration, 3.97 (*SD* = 1.49) on organizational trust (Organizational Trust was measured by a 7-item scale [53] (1 = strongly disagree, 7 = strongly agree)), 3.73 (*SD* = 1.58) on organizational justice (Organizational Justice was measured by a 6-item scale [54] (1 = strongly disagree, 7 = strongly agree)), 3.11 (*SD* = 1.49) on psychological distress (Psychological Distress was measured by a 4-item scale [55] (1 = never, 7 = always)) and 2.61 (*SD* = .77) on physical symptoms (Physical Symptoms were measured by the 12-item version [56] (1 = never, 7 = always)).

### 3.2. Measures

We measured sleep duration, sleep quality, PA, and being bullied as antecedents to WB perpetration. WB Perpetration, being bullied, and sleep quality data were measured weekly via mini-surveys. Sleep duration data were collected daily by fitness trackers. We aggregated daily data to form weekend and weekday average sleep duration. Physical activity data, as steps taken, were also collected daily by fitness trackers. We aggregated daily data to form weekend and weekday average steps data. We controlled for baseline measures of supervisory position, psychological distress and mental illnesses.

WB Target: The four-item EAPA-T-R scale [52] developed for diary studies was used (e.g., My correspondence, telephone calls, or work assignments have been controlled or blocked; 1 = never, 7 = daily). The scale was developed in Spanish and had an English version. The questions in English were translated into Turkish using the translation back procedure. We checked the scale’s validity with a sample of 302 responses in our first wave of LS. The correlation coefficients of all questions in the EAPA-T-R WB Target scale (Q1 *r* = .69, Q2 *r* = .75, Q3 *r* = .72, Q4 *r* = .64) against the total score were higher than the critical value of .113 (300 df (.05)) for a sample size of 302. Therefore, the target scale was valid. The internal consistency for the scale was also acceptable within our sample (Cronbach’s α = .74) [57] during the first wave of the LS.

WB Perpetration: The Spanish four-item EAPA-T-R scale [52] was converted to an active style to measure how frequently participants directed each of the four behaviors towards others during the last week [52] (e.g., I controlled or blocked the correspondence, telephone calls, or work assignments of others; 1 = never, 7 = daily). Every week the results of this scale were used to monitor diary study participants’ WB perpetration behavior, frequency, intensity, and duration. We checked the scale’s validity with a sample of 302 responses in our first wave of LS. The correlation coefficients of all questions in the modified EAPA-T-R WB Perpetrator scale (Q1 *r* = .65, Q2 *r* = .74, Q3 *r* = .64, Q4 *r* = .55) against the respective total score were higher than the critical value of .113 (300 df ( .05)) for a sample size of 302. Therefore, the modified WB perpetrator scale showed evidence of its validity, and the internal consistency for the scale was acceptable (Cronbach’s α = .77) during the first wave of the LS. The questions formed in Spanish were translated into Turkish using the same translation back procedure.

We also compared EAPA-T-R target scores, and modified EAPA-T-R perpetration scores with the single-item victim and bully scores obtained at Time 1 of the LS (*n* = 2508). There was a significant correlation between the EAPA-T-R target score with the victim score (*r* = .51, *p* < .01), and the modified EAPA-T-R perpetration score with the bully score (*r* = .58, *p* < .01) at Time 1 of the LS. This result supported the construct validity of the EAPA-T-R target and modified EAPA-T-R perpetration scales. These values were expected since not all employees exposed to bullying feel victimized [52].

Sleep Quality: We verified the sleep quality by a single-item self-rated sleep quality [58,59] measure (e.g., during the last week: How satisfied were you with your sleep quality? 0 = best possible sleep, 10 = worst possible sleep). A single-item measure was previously shown to have acceptable reliability and validity [60,61]. The single-item question was in English and was translated to Spanish and Turkish using the same translation back procedure mentioned above.

Fitbit Inspire 2 (Fitbit LLC, San Francisco, CA, USA) trackers were distributed to the participants to track their sleep duration and step counts. The daily sleep duration and step data were converted to weekly averages to be used in the analysis. Although trackers also reported sleep quality to the users, these measurements were not available for download from the consolidator platform, Fitabase (Small Steps Labs LLC, San Diego, CA, USA).

Sleep Duration: The fitness trackers monitored daily sleep duration and supported the subjective measure of sleep quality [62]. Fitbit devices record significant movements during sleep as sleep interruption or ‘awakening.’ Thus, such moments are recorded as awake, and sleep duration is indicated net of these awakenings. Data were collected on a 24-h per day basis (00:00–23:59). The average weekend sleep duration was computed by averaging Saturday and Sunday sleep (anytime slept between Saturday morning and Monday morning). Similarly, average weekday sleep duration was computed by averaging sleep from Monday to Friday (anytime slept between Monday morning and Saturday morning).

Steps taken: During the diary study, PA was also monitored by fitness trackers in terms of daily steps taken. The device can pick up activity and report steps taken. Data were collected on a 24-h per day basis (00:00–23:59), and data were computed to yield the average weekend steps by averaging Saturday and Sunday (Saturday 00:00–Sunday 23:59). Similarly, the average weekday steps were computed by averaging steps from Monday to Friday (Monday 00:00–Friday 23:59).

Supervisory position, psychological distress, and mental illnesses (see Section 3.1), were chosen as control variables based on the results of the first wave of the LS. A single item question inquired about the supervisory role; “Do you have a supervisory position (do you manage other employees)? Yes = 1, no = 0.

### 3.3. Procedure

As longer-term diary studies help us understand the temporal precedence between individuals’ perceptions of their environment and individual states over time, we designed a 24-working-week diary study to examine the predictors of WB perpetration. We adopted a ‘full panel design’ in which predictors; exposure to bullying was measured weekly, sleep, and PA were measured daily, and the dependent variable, WB perpetration was measured weekly. We stopped data collection if the participants were not working (e.g., on vacation, sick leave, or job change). The study started in May 2021 and lasted until January 2022 (Figure 2).

Before the start of the diary study, we met the participants. We explained that while the study is about employee health, there will be questions about work conditions. We urged them to answer as honestly as possible and told them there were no right or wrong answers. We gave them Fitbit fitness trackers and requested them to wear them around the clock except for three hours per day while they showered or charged the device. We created Email and Fitbit accounts on the participants’ behalf with dummy names and email addresses to guard the confidentiality of the project. We assisted participants in downloading the app on their mobile phones, logging in with the new account information, and synchronizing the device. We regularly checked their data and reminded them to wear, synchronize, and charge their tracker throughout the study duration [63]. To be able to verify the usage, we requested access to heart data as well. When participants did not wear the device, we reminded them to reduce data loss. We regarded the day as missing if there were more than 3 h of not wearing. We kept track of synchronizing at least every two days to reduce the risk of losing data. On Saturdays, daily data for each week was downloaded from the consolidator platform.

Before the diary study started, we obtained informed consent from participants to monitor their data for the stated period. We ensured that personal information (name, age, etc.) would be kept confidential and that personal data would not be shared with any other entity or person outside the investigation. Before the start of the study, we gave participants the task [64] of ‘engaging in good health habits’ (e.g., reduced consumption of alcohol, improved sleep, optimal PA, etc.). Each Friday after 17:00 h, a short questionnaire was sent to participants via personalized links. Every Saturday, participants were reminded to complete the survey by the end of Sunday as it would become unavailable. Weekly assessments reduced retrospective bias [65,66]. To keep motivation high, we followed up with each participant who missed the weekly questionnaire [63]. It was promised that certificates of Wellness Training would be given upon full participation, and Fitbits would be gifted. We checked whether the responses were valid (e.g., the same answers throughout the diary) and found no invalidity.

### 3.4. Strategy of Analysis

The repeated measurement of the participants, where the weeks are nested within persons, made it necessary to perform multilevel analysis on the data with weekly observations at the first level (Level 1; *N* = 720) and persons at the second level (Level 2; *N* = 31). In our multi-level-analyses, we used person-mean centering (group-mean) for weekly antecedents to ensure Level 1 coefficients represent within-person effects. Grand-mean centering was used to understand the relationship between the control variables (psychological distress, supervisory position and mental illness) and WB perpetration on Level 2 [67]. We performed a three-step procedure. First, a null model containing no predictors was tested to see how much of the variance in WB perpetration was within and between levels, where within-level variance was significant. Second, WB perpetration was regressed upon weekly antecedents and without control variables to test for the main effects of weekly antecedents on WB perpetration. As the model fit was acceptable, control variables were introduced on the between-level in the third step to see if the model fit improved. The analysis was conducted in a multilevel format using Mplus 8.8 [68]. The Maximum Likelihood Robust estimator was used for regression analysis which does not provide confidence intervals on the multilevel. In evaluating the adequacy of models, we considered the following fit indices: the chi-square, the comparative fit index (CFI), the Tucker–Lewis index (TLI), the root-mean-square error of approximation (RMSEA) and the Standardized Root Mean Square Residual (SRMR). When evaluating the goodness-of-fit of structural regression models with a chi-square value, a non-significant *p*-value indicates a good fit. Other indices of model fit were also considered in this study. Based on stringent recommendations [69], a CFI and TLI value of .90 or greater indicated a good fit, and values of .95 or greater represented excellent fits. The RMSEA point estimate indicated a good fit to the data at values of .10 or less, with values of .06 representing excellent fits. SRMR value of a maximum of .08 indicates a good fit. As for the missing values, we relied on the Full Information Maximum Likelihood (FIML) method to reduce the response bias [70]. When using FIML, missing values (either by not completing the data collection weekly, not wearing the Fitbit, or by missing just one item or a scale) are not changed or imposed, but they are processed within the analysis model, allowing the use of all the available information to predict the model. We conducted our primary analysis of descriptive statistics, correlations, and independent *t*-tests in SPSS 26 (IBM Corp, New York, NY, USA).

## 4. Results

### 4.1. Descriptive Statistics and Correlations

We conducted a Pearson correlation analysis of variables measured in the first wave of LS to determine the control variables for the diary study (Table 2). We employed this approach since the diary sample size was too small to give reliable results on between-level correlations. As expected, being a supervisor (*r* = .08, *p* < .01), mental illnesses (*r* = .11, *p* < .01), neurotic personality trait (*r* = .11, *p* < .01), psychological distress (*r* = .21, *p* < .01), physical symptoms (*r* = .21, *p* < .01) and being bullied at work (*r* = .42, *p* < .01) were positively correlated with perpetration. Meanwhile, age (*r* = −.10, *p* < .01), agreeableness (*r* = −.16, *p* < .01), and conscientiousness traits (*r* = −.18, *p* < .01) were negatively correlated with perpetration. As for work environment, organizational trust (*r* = −.09, *p* < .01), and organizational justice (*r* = −.08, *p* < .01) at work were negatively correlated with perpetration.

### 4.2. WB Perpetration Frequency, Intensity and Duration

The WB perpetration scores were monitored throughout the study. At the end of the study, we grouped participants into five groups in terms of the intensity, frequency and duration of their perpetration behavior measured by the weeks acted out (Figure 3).

One of the aims of the diary study was to monitor perpetration as a process and observe how it unravels and evolves for each individual. The first three groups (1–3) represent how individuals may be caught up in the process where WB perpetration acts continue despite vacations, holidays or sick leaves with varying intensities. Previously, employees were classified as perpetrators when they performed at least one negative act per week or at least four negative acts per month [71]. Based on this definition, participants who reported one negative act even for one week were included in the “involved” group. Therefore, when classified as per frequency and duration, eight participants reported WB perpetration only for one week, seven reported between 2 and 4 weeks, and sixteen participants reported from 5 weeks (21% of the study period) up to 24 weeks (100% of the study period). Differences in behavior patterns showed us that perpetrators are not one homogenous group, as mentioned by previous studies [72]. While they may differ among themselves in the intensity of the act, WB perpetration behavior for most was sustained for a long time, and individuals did not change their behavior.

### 4.3. Hypothesis Testing

We conducted a series of multilevel confirmatory factor analyses with Mplus 8.8 [68] to discriminate the variables to be included in the study (e.g., being bullied; sleep quality; weekend and weekday sleep duration; weekend and weekday step counts). We assessed if the previous week’s variables were predictors for this week’s perpetration. The model fit was unacceptable. Therefore, we concentrated on this week’s variables in this week’s perpetration incidences. The only combination of predictor variables that had a model fit was sleep quality, weekday steps and target score (χ2 (df = 6) = 6.5, CFI = .98, TLI = .96, RMSEA = .01, SRMR_within_ = .03, SRMR_between_ = .09). These variables explained 5.90% of the variance in perpetration score on the within-level while control variables explained 27.70% of the variance on the between-level. Table 3 presents standardized estimates, standard errors, and t and *p* values for all predictors. Without the control model fit was lower (χ2 (df = 3) = 3.4, CFI = .97, TLI = .93, RMSEA = .01, SRMR_within_ = .03, SRMR_between_ = .00).

Hypothesis 1 stated that weekly exposure to bullying would positively predict weekly WB perpetration. Results supported our hypothesis because target scores reported during the week strongly and positively predicted WB perpetration at the within-level. (β = .23, SE = .07 *p* = .002). Of the 31 participants, 12 (38.71%) were supervisors with a group average perpetration score of 1.41 (1.14 for 19 non-supervisors). All but one were bullied, and their average target score was 1.47 (1.53 for 19 non-supervisors). The Supervisor group reported 129 perpetration incidences and 135 target experiences; six were bullied more times than they bullied others, and the remaining six bullied others equal to or more times than they were bullied. This result shows us that even supervisors could be a target of WB and suffer resource losses. When unable to cope, they show negative behaviors toward others. In the non-supervisor group of nineteen, 127 WB perpetration and 235 target experiences were reported; fifteen of them were bullied more times than they bullied others, and the remaining group of four bullied others equal to or more times than they were bullied. Our results show that bullied employees enter a defensive mode to preserve the self and act out.

Hypothesis 2a stated that steps taken during the week would positively, and Hypothesis 2b stated that during the weekends would negatively predict weekly WB perpetration. The multilevel analysis results on steps supported Hypothesis 2a because steps taken during the week positively predicted WB perpetration at the within-level (*β* = .07, SE = .03 *p* = .04). H2a was confirmed. The weekend steps data did not fit the data. H2b could not be tested. Therefore, our results showed that excess over the individual’s weekly average PA as steps taken during the week is related to weekly WB perpetration.

Hypothesis 3a stated that weekly sleep quality would negatively predict weekly WB perpetration. Average sleep quality for 24 working weeks was 5.98 out of 10 and ranged between 2.96 and 7.96. All participants were asked about their sleep troubles before joining the diary study. They scored an average of 2.98 out of 7 on the Karolinska Sleep Scale (A nine-item short version of the Karolinska sleep scale was used [73] to set the base measure for sleep quality (1 = never, to 7 = all the time), participants were also asked if they had similar sleep difficulties in childhood.) and reported similar sleep difficulties in their childhood, scoring 2.58 out of 7. Despite participants’ poor baseline sleep quality, the relationship between sleep quality and WB perpetration was insignificant at the within-level (*β* = .02, SE = .05 *p* = .65). Therefore, H3a was not supported. We also expected short sleep to cause stress, anger and change in mood, resulting in perpetration behavior. Having adequate sleep is one of the primary resources of COR [22]. The participants recorded an average sleep duration of 6.21 h within 24 weeks. They slept more during the weekends (6.83 h) than on weekdays (6.21 h). Supervisors, on average, slept less (5.91 h) than non-supervisors (6.69 h), but their sleep quality was higher at 6.21 as opposed to 5.84 out of 10 for non-supervisors. However, sleep duration did not fit any models; therefore, H3b could not be tested. We also tested if sleep moderates the relationship between being bullied and bullying others, but the results were inconclusive as the model did not fit the data.

While being bullied, sleep and PA were tested as antecedents to perpetration on a within-level; on a between-level, we controlled for psychological distress, supervisory position and mental illnesses.

The participants reported an average psychological distress score of 3.11 (*SD* =1.49) compared to the rest of the LS participants of 2.39 (*SD* = 1.49), which was a significant difference, *t* (2429) = 3.61, *p* < .00. These results suggested that our sample may have been confronted with especially stressful events. As per COR, events are considered stressful if the stimulus leads to emotional upset, psychological distress, or physical impairment [74]. At the baseline, they had a higher target score (*M* = 2.28, *SD* = .86, *t* (2384) = 3.84, *p* < .00), higher physical symptoms (*M* = 2.61, *SD* = .77, *t* (2404) = 1.92, *p* = .05), lower organizational trust (*M* = 3.97, *SD* = 1.49, *t* (2483) = −3.82, *p* < .00) and lower organizational justice environment (*M* = 3.73, *SD* = 1.58, *t* (2477) = −3.73, *p* < .00), which were significantly different from the rest of the LS group. Our results showed that psychological distress predicted perpetration as a control variable on the between-level. Our sample had twelve supervisors (38.71%), and the supervisory position was also predictive of perpetration in between-level analysis. Lastly, we controlled for mental illnesses. Nine (29.03%) out of 31 participants have been diagnosed with mental illnesses (Depression = 8, Obsessive-compulsive disorder = 1), and mental illness score predicted lower WB perpetration.

In summary, our results showed that individuals engaged more in WB perpetration towards others during the weeks they were bullied and when they were more physically active than their average levels. The group engaged in increased perpetration if they were supervisors and reported higher psychological distress. The reports of WB perpetration were lower for participants with mental illnesses.

## 5. Discussion

The present study utilized a within-person approach, incorporated the perspectives of perpetrators, and focused on the dynamic process of WB. We believe we have made two important contributions to the literature on WB from the perspective of the perpetrators.

First, by applying a 24-week diary design, we demonstrate the dynamics of perpetration incidences on a within-level and how they emerge, intensify, prevail or die away. Our results support the theoretical structure of COR. Being bullied tarnishes major resources of the COR (e.g., feelings of being valuable and successful). Once employees lose these resources, stress, anger, and depressive moods occur. Loss of self-esteem may be met with attempts to re-establish self-esteem or losses incurred in some conflicts may be compensated by gains in others [22]. Employees with no supervisory roles usually have less power, are more exposed to WB and may lack experience handling work conflicts. However, we also showed that supervisors who bully are also being bullied. The positive correlation between active bullying and being bullied has been confirmed in previous cross-sectional empirical research [3,5] and reviews [75,76] on a between-level. Now we add daily and weekly diary results measured on a within-level. Previous research also indicated that employees show continuous linear responses to being bullied and bully others [3]. Based on the frequent, long-lasting nature of the acts, our results align with the previous result that workplace bullying differs from interpersonal conflicts in duration and frequency [77]. Our results also support the three-way model [20], suggesting that perpetration behavior may arise due to unsolved escalating interpersonal conflicts, ineffective coping and a toxic work environment. Furthermore, our participants scored high in psychological distress, were caught up in a vicious cycle of workplace bullying and had lower-than-average organizational trust and justice environments.

The second major contribution is incorporating psychological and, more importantly, physical health into the study of perpetrators using objective data and not only using perceptions. To the best of our knowledge, in previous studies, perpetrators were not assessed for their stress levels and mental and physical health. Our results showed that excess PA during the week impacts the individual reporting weekly WB perpetration. This result suggests that excess PA at work, in the household, during leisure time or commuting to work may exhaust the individual. There may be several explanations for physical fatigue. Firstly, during the research, 29% of the participants switched to working from home to the office, and 19% changed jobs and positions, which may have caused lifestyle and commuting schedule changes that may have drained their resources. Second, except for one, all our participants were being bullied. They may have relied on physical activities during the weekdays as a stress relief tool to recuperate lost resources due to being bullied. However, these activities may have caused physical fatigue along with weekday tasks in other domains, diminishing health and stress resistance capacity instead of enhancing it. Third, participants may have exerted too much effort staying active during the weekdays. Previous research with activity trackers has shown that steps taken may increase throughout the study since the participants know they are being monitored [78]. As we hypothesized that PA may trigger WB perpetration incidences, we encouraged the participants to engage in good health habits during the study as much as possible. We told them that engaging in the task is up to them. We did not mention that they should walk more or increase their average steps. Despite this, some participants may have forced themselves to increase activity levels to adhere to the “good health habits” task. If participants engaged in irregular PA, these activities may have also exhausted them, facilitating increased perpetration incidences. We also tested if PA moderates the relationship between being bullied and bullying others, but the results were inconclusive as the model did not fit the data. Another physical health marker was sleep (i.e., quality and duration). While we could not test sleep quantity, our results showed that variations from average weekly sleep quality did not predict perpetration, diverging from the positive link between sleep quality and various aggression constructs established by previous studies [43,46].

Thirdly, our control variables for between-level showed that reports of WB perpetration tended to arise more from supervisors, from participants with high psychological distress and without mental illnesses. Many victims are being bullied by their supervisors, but supervisors’ chances of being reprimanded, dismissed, or socially isolated are low, so they may also perceive a low risk for themselves. Supervisors may also use bullying as a disciplinary action or to eliminate unwanted employees [79]. COR can explain supervisors’ behavior. They weigh potential resource gains (e.g., team success) against resource losses (e.g., time spent supporting subordinates). When their resources are low, supervisors may become defensive in their resource investment strategies and thus trigger resource losses around themselves [24]. Previous research showed that mental illnesses were positively correlated with WB perpetration [10]. However, our results showed that mental illnesses predicted lower perpetration. COR suggests severe resource losses and resource investments that fail to resolve conflicts are responsible for depression [74]. Based on this, we may say that the participants with mental illnesses may already be low on resources. They might feel that their health, well-being and employment are at risk. Experiencing bullying may put additional stress on this group. By weighing the outcome potential instead of further risking their resources, they may refrain from engaging in conflicts and hence perpetration behavior is less prominent.

### 5.1. Theoretical and Practical Implications

We contributed to COR with two important pieces of evidence. Firstly, we demonstrated an empirical example of resource loss spirals (Figure 2). Resource loss spirals emerge when an individual with low resources attempts to redeem the self by using remaining resources to cope. This effort results in futile coping without positive outcomes but with higher distress [24]. During our study, the frequency of being bullied was 370 observations; the WB perpetration frequency was 256 observations of behaviors; 80% (206) of the total perpetration behaviors appeared during the week of participants being bullied. For seven participants, the resource loss spirals of bullying and perpetration continued between 18–24 weeks, suggesting that aggressive responses did not work, conflicts were not resolved, and resource loss spirals continued. Previous research showed that once perpetration behavior starts, individuals would tend to resist change [71].

Secondly, we contributed to COR on psychological and physical health, which WB scholars usually do not cover. COR suggests that when individuals invest in resource gains, and if this investment does not provide a good return, it may be perceived as a loss of expected gains and create stress [74]. As the participants were in a wellness program, they may have invested time in PA, expecting to feel healthier. If these expectations did not materialize, they may have perceived the result as a resource loss and displayed negative behaviors to regain resources. Similarly, engaging in excess weekday and work-time PA may also physically drain employees, resulting in resource losses. COR also suggests that individuals with low resources are more prone to resource losses [22]. If employees were low on resources due to a stressful work environment (e.g., being bullied), excess PA may also have caused further resource losses. These losses may have been in the form of physical or emotional fatigue. Therefore, the resource losses caused by (i) unmet expectations, (ii) excess PA, and (iii) a toxic work environment may have pressured employees to display coping behavior and protect the self through perpetration behavior.

Employers are primarily responsible for protecting employees from bullying that undermines their reputation, health, and dignity. Therefore, top management guided by human resources departments should create just and fair policies, publish the organizational code of conduct to guide the employees on unacceptable behavior and hold employees accountable in case of breach [80]. They also should make sure that job roles are clear, workloads are manageable, and conflicts are actively managed to inhibit the emergence of bullying. This study’s results suggest that organizations must be alert to their employees’ physical and psychological well-being. By mass surveying or through their occupational physicians, organizations may ensure that the PA employees undertake during weekdays are not over and beyond their endurance levels. Organizations may engage in job redesigns to enhance employee health and wellness and look beyond the working hours to support and coach employees against physical strain.

### 5.2. Strengths, Limitations and Future Research

The first strength of this study was the 24 working week design analyzing the dynamics of perpetration behavior, which was long called for [75]. Our study was one of the first to observe active and potential perpetrators over and above one month and analyze within-person fluctuations. Therefore, by monitoring individuals’ lives for over nine months with 24 waves, we were able to maximize our chances of detecting patterns in their perpetration behavior related to the variables studied [81]. Second, the weekly diary design reduced the risk of retrospective bias as opposed to looking back six months, as generally conducted in cross-sectional or longitudinal studies. Third, we gathered sleep duration and step count directly from the fitness tracker app through an aggregator platform, increasing the objectivity of the data and reducing the reliance on self-reports.

The sample size was one of our main limitations. Despite the relatively small sample size, we had an appropriate number of observations for within-level analysis (*N* = 720 observations). However, our sample size was not appropriate for between-level analysis as it was less than the suggested 50 for unbiased estimates on the between-level [60,82]. Our second major limitation was that our results for the relationship between predictors and WB perpetration remain correlational and not causal. Although we tested if antecedents predicted WB perpetration with one week lag, the model fit was unacceptable. Perhaps the reason for this was the breaks given during the 24-work week study. A general limitation was the problem of common method variance due to self-reports [83]. To mitigate this risk, (i) we separated the measurement of the predictor variable from the outcome variable by announcing the study as a wellness training program, (ii) assured respondents that there are no right or wrong answers and that they should answer questions as honestly as possible, and (iii) we changed the order of questions in the scales and the order of scales. Another limitation was the social desirability effect, where participants may have underestimated their responses, particularly on perpetration incidences. To mitigate this risk, (i) we assured anonymity, (ii) conducted the research outside participants’ organizations, (iii) did not inquire about where they worked, and (iv) we did not mention “bullying” throughout the study. We believe this risk was largely mitigated, evidenced by the persistent reports on WB perpetration except for one-time perpetrators in the study. One may also see the low frequency and duration of perpetration acts in Groups 4 and 5 (Figure 2) as a study limitation and argue that low-intensity perpetration groups should not have been classified as WB perpetrators. However, we examined perpetration weekly rather than seeking accumulated scores (e.g., the previous six months). Therefore, we used every observation possible. Finally, Fitbit activity trackers may not provide accurate measurements for sleep and PA [84]. However, group-mean centering mitigated this risk using relative values rather than absolute numbers in the within-level analysis.

In the future, WB perpetration studies may focus on health markers such as diabetes, mental illnesses, heart rate and conditions, sexual health, menstruation, maternity, menopause and health practices such as meditation, yoga, or mindfulness exercises. As workplaces are important places of activity for promoting public health and well-being initiatives, WB perpetration research may be teaming up with occupational health physicians or union representatives. In the future, fitness trackers, smart watches, diabetes patches and wellness apps may be used to track the relationship of health with perpetration. Studies with fitness trackers on physical activity may incorporate surveys to enhance the knowledge of data collected (e.g., classifying collected data as energizing or draining). Future studies testing causality between perpetration and predictors may use more frequent or uninterrupted data collection cycles and add other sources of information through subordinates or colleagues.

## 6. Conclusions

This study examined the impact of sleep, PA and being bullied on employees’ within-level perpetration behavior using a multi-source 24 work week diary study. Our results showed that on a within-level, employees who were physically active more and were being bullied more had higher perpetration behavior, while sleep quality was not related to perpetration behavior. On a between-level, supervisors more than non-supervisors and employees who are high on psychological distress showed higher perpetration behavior. In contrast, employees who had mental illnesses exhibited lower perpetration acts compared to mentally healthy employees. Our results aligned with the theoretical notions presented in the COR and three-way model. This daily and weekly diary study extended previous research that perpetration behavior is not a constant phenomenon but varies systematically and unsystematically. Future studies should investigate what causes perpetration and such fluctuations [71]. Our findings broadened the conceptual view of what may cause perpetration by focusing on the physical health of perpetrators as well as their psychological health. We urge organizations to implement measures to fight bullying and refrain from creating new victims as they may become new perpetrators. To reduce WB prevalence, we also urge organizations, scholars and practitioners to listen, understand, and help perpetrators change their behaviors. We hope this research will inspire future researchers to adopt a more dynamic way of thinking about the complexities of perpetration behavior.

## Figures and Tables

**Figure 1 ijerph-20-00479-f001:**
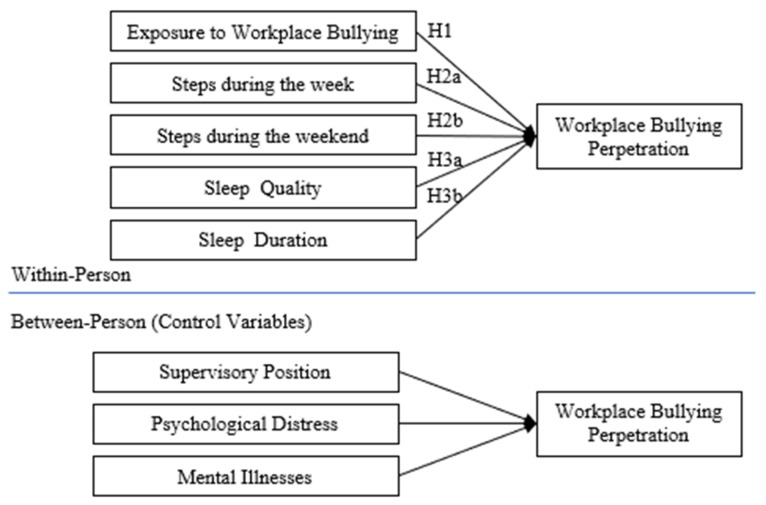
Hypothesized relationships among study variables.

**Figure 2 ijerph-20-00479-f002:**
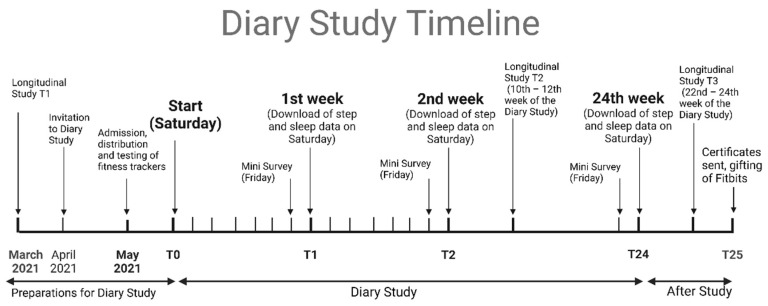
Diary Study Timeline (Created by Biorender.com).

**Figure 3 ijerph-20-00479-f003:**
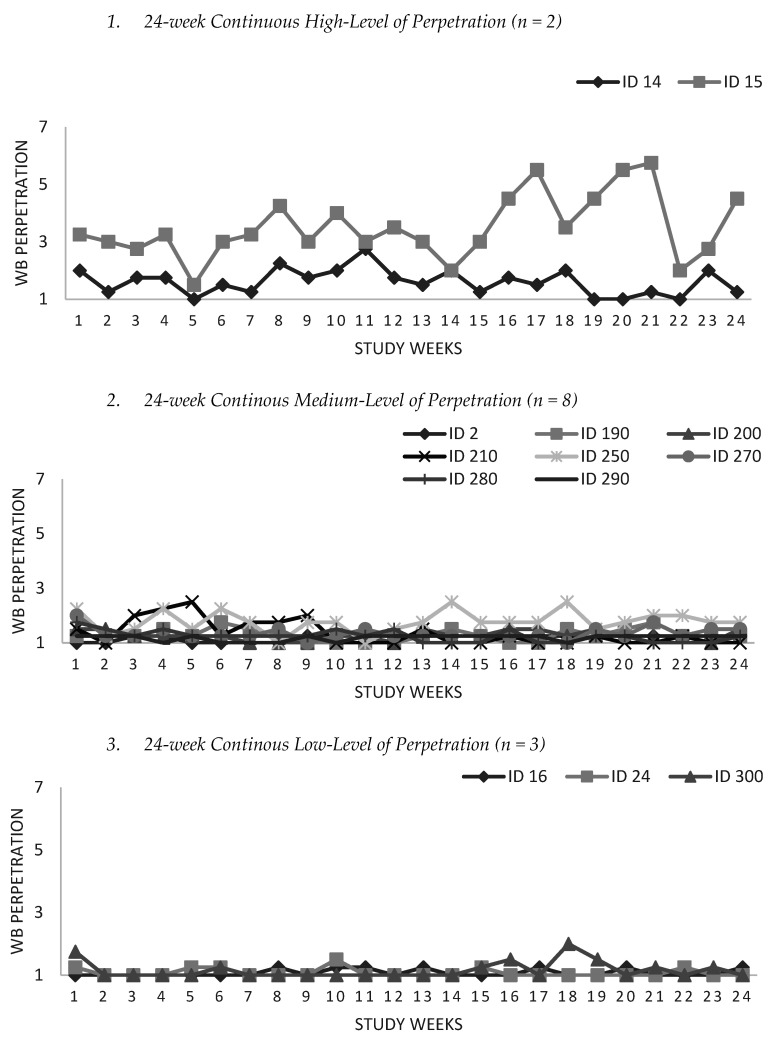
Groups in Diary Study by perpetration frequency, intensity and duration.

**Table 1 ijerph-20-00479-t001:** Sociodemographic Characteristics of Participants at Baseline.

	Participants			Participants	
Number of Participants	31			31	
Baseline Characteristics	*Mean*	*SD*	Baseline Characteristics	*n*	%
Age	37.94	12.27	Organization Sex Ratio		
Tenure	6.39	8.24	Female-dominated	13	41.94%
Work Days	5.10	.70	Male-dominated	14	45.16%
Extraversion	4.05	1.17	Balanced	4	12.90%
Agreeableness	5.65	.83	Supervisor		
Conscientiousness	5.21	1.15	Yes	12	38.71%
Neuroticism	4.35	1.30	No	19	61.29%
Intelligence and Imagination	4.92	1.14	Living in		
T1 Victim Score	2.16	1.70	Turkey	18	58.06%
T1 Bully Score	1.32	1.14	Spain	13	41.94%
T1Target Score	2.28	.86	Gender		
T1Perpetration Score	1.30	.39	Female	15	48.40%
T1Organizational Trust	3.97	1.49	Male	16	51.61%
T1Organizational Justice	3.73	1.58	Mental Illness		
T1Psychological Distress	3.11	1.49	Yes	9	29.03%
T1Physical Symptoms	2.61	.77	No	22	70.97%

**Table 2 ijerph-20-00479-t002:** Correlations of main variables in the longitudinal study (LS).

Variables		*M*	*SD*	1	2	3	4	5	6	7	8
1	Age	34.46	12.25	-							
2	Gender	.43	.49	−.02	-						
3	Supervisor	.28	.45	.21 **	.17 **	-					
4	Mental Illness	.13	.34	−.00	−.09 **	−.02	-				
5	Extraversion	4.26	1.14	−.08 **	−.00	.08 **	−.00	-			
6	Agreeableness	5.52	1.05	.13 **	−.20 **	.01	−.02	.18 **	-		
7	Conscientiousness	5.43	1.17	.16 **	−.03	.06 **	−.10 **	.08 **	.20 **	-	
8	Neuroticism	3.74	1.15	−.15 **	−.26 **	−.08 **	.24 **	−.14 **	−.03	−.23 **	-
9	Intelligence/Imagination	5.02	1.16	−.16 **	.02	−.00	−.02	.22 **	.23 **	.08 **	−.09 **
10	T1 Organizational Trust	4.84	1.28	−.05 *	.05 *	.05 *	−.16 **	.12 **	.14 **	.10 **	−.21 **
11	T1 Organizational Justice	4.66	1.41	−.04	.06 **	.08 **	−.16 **	.11 **	.10 **	.03	−.21 **
12	T1 Psychological Distress	2.40	1.11	−.15 **	−.12 **	−.00	.25 **	−.10 **	−.05 *	−.11 **	.40 **
13	T1 Physical Symptoms	2.33	.83	−.20 **	−.21 **	−.05 *	.27 **	−.03	−.02	−.10 **	.38 **
14	T1 Target	1.66	.91	−.07 **	−.06 **	.01	.18 **	−.01	−.11 **	−.11 **	.20 **
15	T1 Perpetration	1.22	.57	−.10 **	.01	.08 **	.11 **	−.00	−.16 **	−.18 **	.11 **
**Variables**		** *M* **	** *SD* **	**9**	**10**	**11**	**12**	**13**	**14**	**15**	
9	Intelligence/Imagination	5.02	1.16	-							
10	T1 Organizational Trust	4.84	1.28	.08 **	-						
11	T1 Organizational Justice	4.66	1.41	.05 *	.80 **	-					
12	T1 Psychological Distress	2.40	1.11	−.01	−.41 **	−.44 **	-				
13	T1 Physical Symptoms	2.33	.83	−.01	−.27 **	−.29 **	.54 **	-			
14	T1 Target	1.66	.91	−.02	−.39 **	−.40 **	.45 **	.35 **	-		
15	T1 Perpetration	1.22	.57	−.11 **	−.09 **	−.08 **	.21 **	.21 **	.42 **	-	

Note: N = 2508; Gender: 0 = Female 1 = Male; Supervisory role: 0 = no; 1 = yes; Mental Illness: 0 = no; 1 = yes; * *p* < .05; ** *p* < .01.

**Table 3 ijerph-20-00479-t003:** Multilevel associations between predictors and WB perpetration (same week).

		WB Perpetration			
		*Estimate*	*SE*	*t*	*p*
Within-level					
	Sleep Quality	.02	.05	.46	.65
	Average Weekday Steps	.07	.03	2.05	.04
	Target Score	.23	.07	3.17	.00
Between-level					
	Mental Illness	−.24	.10	−2.42	.02
	Supervisory Role	.28	.11	2.60	.01
	Psychological Distress	.37	.19	1.99	.05

Note: *n* = 31 participants, *N* = 720 measurement occasions.

## Data Availability

The data used in this manuscript can be found on the Open Science Framework using the following link: https://osf.io/q2jyd/ (accessed on 22 December 2020).

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
