# Peer review of "A Matter of Health? A 24-Week Daily and Weekly Diary Study on Workplace Bullying Perpetrators’ Psychological and Physical Health"

_ijerph, 2022, doi:10.3390/ijerph20010479_

Round 1

Reviewer 1 Report

Dear Authors

Thank you for the opportunity to review your manuscript which considers workplace bullying with a focus on the perpetrator.

There has been few papers that consider the risk factors related to the perpetrator so this paper offers some interesting insights.

The Abstract describes the paper quite well.

Introduction: This section is quite wordy and could be reduced to focus in on the hypotheses that are being addressed, while ensuring that the many different variables that are to be included in the methods are introduced and justified here.

The reader is left with a series of hypotheses and a final and clear statement of aims at the endo of the introduction would be helpful.

Methods: The methods section is difficult to follow. The initial statement is that there are three waves, yet each wave is unclear.

Students are used to recruit for one wave and then waves 2 and 3 follow, but it was unclear how these waves were run and whether all participants that participated in wave 1 were included in the other waves, nor how the 38 participants were actually chosen from those in the survey i.e. the selection for the diary participants was inadequately defined.

During the description of the methods, the concept of neuroticism and other factors were introduced, yet these were not included in the initial outline of workplace bullying in the introduction.

It is difficult to follow which scales were included in which part of the study. This in important when understanding the results.

The EAPA-T-R scale used to measure the experience of bullying seems to have been used with some revisions to measure the perpetration of bullying. While there is some logic to this, it is unclear whether this has been done before and validated in any way. It is important to the understanding of the validity of the study to ensure that the definition of perpetration is adequate.

The recruitment of the participants via ‘students who get course credit for identifying participants’ and the presentation of the study as a ‘deception study’ seems a little problematic. These issues would be assessed by the ethics committees. Therefore, this study should have had robust ethics approval from an accredited organisation in both Spain and Turkey. Similarly, there would need to be ethics for all the waves of the research and for the diary collection. Currently, the names of these ethics committees are not available, and there are no ethics numbers available to enable validation of the ethics process.  Before this study is accepted for publication, this issue would need to be addressed as this information needs to be available for transparency.

It is unclear how the authors addressed the potential for the study to cause psychological distress experienced by the participants when responding to the questions about personality, health, sleep and workplace experiences. This should be included to demonstrate that best practice approach to research was undertaken.

In the measures section it is not clear what scale is used for measuring neuroticism and other factors that are reported in the results table. It is unclear how mental illness was measured – was it simply the statement about the physician diagnosing ‘mental illness’ or were specific examples of what ‘mental illness’ might be included to assist the respondent. Otherwise, this question seems problematic unless there is some way to justify this. Again, if the authors wish to include mental illness in the method section, the statement about it being related to perpetration of workplace bullying belongs in the introduction, rather than the methods.

In general, it is unclear how many of the items recorded in Table 1 are derived. The issue of sleep quality comes into the paper multiple times and seems to have been measured using the Fitbit and the survey, they later in the paper the authors suggest that only the survey was used for an evaluation of sleep quality, even though it seems that the authors had the full Fitbit data.

It is unclear whether the larger survey was anonymous if the survey was not anonymous, then did that fact impact on the results?

In general, the references in the methods sections should ones used to justify the methods used rather than explaining the issues associated with perpetration of bullying.

There is a statement that “We stopped data collection if the participants were not working”, yet it is unclear if this was a temporary situation just while the participant was on holidays and whether they then re-started on return to work.

Overall, the methods will need to be reviewed with the lens of the reader to ensure that adequate detail is included to ensure that the study could be readily replicated.

The results include Table 1, with the diary participants being compared to the Time 1 survey.

It is unclear why this is being presented. Why is it important that the diary participants are the same, or not the same, as the larger survey. The rationale for this and for other items in the table needs to be stated in the results in words (succinctly), especially since “diary sample size was too small to give reliable results on group-level correlations”.

In 4.2, the authors describe the “Perpetration scale” yet that term is not included in the Methods. The information in the methods should enable to reader to expect to see the graphs that are being provided in the results section. While these are interesting, the methods do not provide enough detail for the reader to be prepared for how the results are being collated.

The graphical representations are quite interesting but there needs to be a short paragraph of interpretation as well.

The use of analysis for within level and between level are not explained in the method – in the analysis section and the use of the term Level seems to refer to something else in the methods.

There are many times within the results and discussion sections where the quality of English makes it difficult for the reader to follow the intent of the sentence. e.g. “we concentrated on this week’s variables in this week’s perpetration incidences.” Reviewing the paper and especially the results and discussion sections to fix these places would greatly benefit the manuscript as a whole.

Statements are made such as the authors stating that they controlled for supervisory position, mental illnesses and psychological distress and other items.  It is unclear how these were controlled for. Having a statistician review of these statistics would be helpful. It would seem that confidence intervals should be available to interpret the results. Again having more clarity in the words of the results as to what the authors found through this aspect of the study would be helpful.

The results section ends abruptly and then moves on to the discussion. A statement at the end would be appropriate to enable a transition.

Discussion: The discussion includes a large amount of information about workplace bullying that was not being measured within this study. The discussion needs to be revised and focused in on the study’s results. The findings about workplace perpetrators of bullying are important and especially identifying the fact that the perpetrators can also be suffering from bullying. These findings seem lost at present in the sea of other facts, many of which no not related to the study itself.

The discussion seems to focus on the diary group, and the results of the survey are left hanging without explaining the meaning to the reader. Meanwhile the discussion introduces many interesting details about bullying that this study does not address which should not be included here. e.g.“Social desirability may have reduced the likelihood of obtaining accurate responses, particularly on perpetration incidences” It is not clear that this has been a finding of the study.

Similarly when the authors state: “Our results also apply to the Three-way model” – it would be better to say “Our results also support the Three-way model”.

The authors appear to have dabbled in significant speculation such as describing the possibility of cognitive dissonance. Also the authors say that their “results show just how quickly bullying creates perpetrators”. It is difficult to draw such a definitive conclusion about this given the low number of participants. I think the authors should qualify this statement e.g. suggesting that the results support the idea that the experience of being bullied can result in that person becoming a perpetrator quite quickly after this.

While a perspectives paper may allow such speculation, this discussion section does not seem the right place for this.

The excessive detail in the discussion is one reason why the reference list is so very long. This includes the theoretical and practical implications. While the speculation is interesting, this section should be an extension of the results rather than a dissertation.

In the limitations, it would be reasonable to highlight the small numbers who participated and recognise the fact that correlation does not necessarily indicate causation so the results of this study should not be over-interpreted. While these statements of limitations are spread across the body of the paper, they are not stated clearly enough in this section of the paper. Stating “Our study is not without limitations which will be discussed together with implications for future research.” In not enough. These statements about the limitations should be included here.

The Conclusion is reasonable,

I hope these comments are helpful.

Author Response

Please see the attachment for answers to your comments.

Thank you very much for taking the time to review our work in such a detailed manner. We appreciate it very much. 

Best Regards

Happy New Year

Reviewer 2 Report

Thank you very much for giving me the opportunity to review your manuscript. I really appreciate the elaborate and thorough design as well as the highly relevant research topic. However, I have some major issues which I would like to describe in the following in more detail:

Hypotheses:

·         I cannot fully follow your derivation of H2a and H2b. Please make your argumentation more clear.

·         Please better differentiate between sleep quality and sleep duration in your derivation of H3a and H3b.

Analyses/Hypotheses:

·         I am not sure whether I understood your analysis procedure correctly: You calculated multiple models, checked which one fitted best and then formulated your hypotheses accordingly? Hypotheses should be formulated prior to analyses. Unless, they do not make sense anymore. Please clarify.

Interpretation of results:

·         Discussion: “First, by applying a 24-weekly diary design, we demonstrate the dynamics of perpetration incidences and how they emerge, intensify, prevail or die away” -> Do your results really say anything about the dynamics of WB? I really appreciate your design, however, the analyses do not say anything about change trajectories, do they? I only can see WB patterns in Figure “Perpetration frequency, intensity and duration”. But what for do you use these patterns in your analyses? Please clarify.

Here are some minor issues as I went through the manuscript:

Abstract, l. 14-16: Revise sentences to reduce redundancy

p. 1, l. 41: all negative behaviors have proven to be stressful. -> which behaviors?

p. 2, l. 55: and on society as a whole -> what does that mean? Can you give an example?

p. 2, l. 79: multiple causes emerged -> for example?

Figure 1: structuring the model from the left to the right would increase readability. Please place all predictors at the left and the outcome on the right. Can you also visualize the different levels the variables refer to?

p. 4, l. 167: „that morning childcare-related and commute-related demands predicted start-of-work-time fatigue“ -> in which way are morning childcare demands and cummute-related demands related to physical activity. Please make clear, because in the paragraph, you focus on physical activity.

p. 4, l. 183: „…and drain resources leading to perpetration“ -> I do not understand this sentence grammatically, please refine.

p. 4, l. 183-186: „and (2) excess over the average physical activity if voluntary (e.g., Leisure time physical activity) would energize the individual, provide coping resources against work stress, feelings of good health and lead to perpetration at work“ -> I cannot follow your argumentation here. Why should feelings of good health lead to perpetration at work?

p. 5, l. 201: „Sleep effects on human behavior were studied under various constructs at work“ -> please rework this sentence grammatically.

p. 5, l. 205: „…that sleep predicts abusive supervision…“ -> which quality of sleep predicts these outcomes? Please be precise in your writing.

The diary study timeline should be named “Figure 2” (Figure 1 is your research model).

Figure „Diary Study Timeline“: Where are the two remaining waves of your longitudinal study (T2, T3)?

Methods: reading the section “3.1 participants” it does not become fully clear the reader how the longitudinal study and the dairy study were related and used in this paper. Please make clear.

Methods, 3.1: what was the full sample size of the longitudinal study, from which you selected 493 people? Please be more precise on the cut-off values you used to select your participants for the diary study.

p. 6, l. 294: „On Fridays, daily data for each week was downloaded from the consolidator platform“, however, in the figure „diary study timeline“ you visualize that data were downloaded on Saturday? Please clarify.

p. 7, l. 313: I know the translation-back translation-procedure, but what is a translation back protocol? Do you have a reference?

p. 8, l. 349-359: I think, here you can shorten some contents.

Özer et al. (2021) is not listed in the reference list. Maybe it should be Özer et al. (2022).

Table 1, sociodemographic characteristics: there are variables listed which are not explained in the measurement section. Please clarify.

Figure “Perpetration frequency, intensity and duration” -> please explain in a note what ID means (participants)

Table 3: please specify, to which time period the variables refer (e.g., this week’s perpetration?)

p. 15, l. 519: I think, if you discuss your results in the discussion with regard to the three-way model (Baillien et al., 2009), you should introduce this model in your introduction.

Discussion: I suggest to substantially shorten the discussion section to make it more concise and to the point.

Author Response

(The authors gave the same response as above.)

Reviewer 3 Report

The paper focuses on a topic of important clinical, social, and political relevance, due to the high prevalence of bullying and its negative effects on mental and physical health, along with its negative impact on the quality of the relationships established in the workplace and occupational health. Moreover, this study contributes to fill the gap concerning the relationship between physical activity of WB perpetrators, as well as to further understanding the impact of sleep quality and duration on the engagement on these violent behaviors in the workplace.

Regarding its structure and content, the manuscript presents a consistent theoretical framework and methodology. Nevertheless, some issues require revision.

In the abstract, the professional occupation and work sectors and age range of the participants is missing, along with its mean and standard deviation.

In what concerns the theoretical framework and hypotheses, I suggest the authors to revise the 2a and 2b hypotheses, as it is not common to expect to observe a positive and a negative effect at the same time. I understand the reasons the authors assumed that excessive physical activity may lead to exhaustion and, consequently, to difficulties in emotional and behavioral regulation. I also acknowledge that studies on the association of physical activity and WB perpetration do not exist. However, most studies cited in the theoretical framework pinpoint the positive effects of physical activity on the employees’ health, well-being, commitment, and engagement with work. Therefore, it possibly would be coherent to assume that this predictive effect is positive, even though the results do not support this hypothesis, and properly address this in the discussion.

In the Method section, information on the description of the sociodemographic characteristics of the participants is lacking, probably because it is included in the Results’ section. This should be revised. In addition, the procedure usually appears after the Measures’ description, and not before. Also, the information concerning the description of the instruments and measures should be included in the Measures’ subsection. Furthermore, it is not clear what are the dimensions assessed by the weekly questionnaire the participants were asked to complete.

Also in the Method, the first sentence of the Measures’ subsection, as it refers to the data collection’s strategy, must be integrated in the procedure.

Author Response

Please see the attachment for answers to your comments.

Thank you very much for taking the time to review our work. We appreciate it very much. 

Best Regards

Happy New Year

Round 2

Reviewer 2 Report

Thank you very much for carefully revising your manuscript and considering my comments, which I hope were helpful for you. I think, the quality of the manuscript substantially improved. I only have few minor issues:

l. 509: “This result suggests that excess PA at work, commuting to work, in the household, or in leisure time may exhaust the individual.” -> please revise: do you mean PA in the household and PA in leisure time?

l. 513: “which may have caused lifestyle and commuting schedule changes that may have been drained their resources” -> please check grammatically

l. 538: “against the resource losses (e.g., by time spent in supporting them) to be incurred” -> I do not fully understand this sentence logically.

l. 566-568: Does that mean that your study increased WB? Maybe you could address this aspect from an ethical perspective.

 I wish you all the best for your future work on the topic.

Author Response

Dear Reviewer 

Thank you very much for your comments. Our answers are attached. 

Best Regards & Merry Christmas
